# Chasing Intracellular Zika Virus Using Proteomics

**DOI:** 10.3390/v11090878

**Published:** 2019-09-19

**Authors:** Pietro Scaturro, Anna Lena Kastner, Andreas Pichlmair

**Affiliations:** 1School of Medicine, Institute of Virology, Technical University of Munich, Schneckenburgerstr. 8, 81675 Munich, Germany; lena.kastner@tum.de; 2German Center for Infection Research (DZIF), Munich Partner Site, 81675 Munich, Germany

**Keywords:** flaviviruses, Zika virus, proteomics, interactome, AP-LC-MS/MS, phosphoproteomics, Label-free Quatification, arboviruses, DENV, WNV

## Abstract

Flaviviruses are the most medically relevant group of arboviruses causing a wide range of diseases in humans and are associated with high mortality and morbidity, as such posing a major health concern. Viruses belonging to this family can be endemic (e.g., dengue virus), but can also cause fulminant outbreaks (e.g., West Nile virus, Japanese encephalitis virus and Zika virus). Intense research efforts in the past decades uncovered shared fundamental strategies used by flaviviruses to successfully replicate in their respective hosts. However, the distinct features contributing to the specific host and tissue tropism as well as the pathological outcomes unique to each individual flavivirus are still largely elusive. The profound footprint of individual viruses on their respective hosts can be investigated using novel technologies in the field of proteomics that have rapidly developed over the last decade. An unprecedented sensitivity and throughput of mass spectrometers, combined with the development of new sample preparation and bioinformatics analysis methods, have made the systematic investigation of virus–host interactions possible. Furthermore, the ability to assess dynamic alterations in protein abundances, protein turnover rates and post-translational modifications occurring in infected cells now offer the unique possibility to unravel complex viral perturbations induced in the infected host. In this review, we discuss the most recent contributions of mass spectrometry–based proteomic approaches in flavivirus biology with a special focus on Zika virus, and their basic and translational potential and implications in understanding and characterizing host responses to arboviral infections.

## 1. Introduction

Viruses are highly adapted intracellular pathogens that co-evolved with their hosts over millions of years. This resulted in a sophisticated adaptation that allowed viruses to engross cellular machineries for their own replication, with the benefit to encode their genetic information with high efficiency. However, this strict dependency on cellular factors comes with the obligation to replicate intracellularly and the requirement of intimate interactions between viral and cellular proteins. Moreover, most pathogens are preferentially replicating in specific organs and cause virus-associated pathologies. Technological developments in the fields of genomics and proteomics became available in recent years, allowing to systematically map interactions between viruses and their hosts in an unbiased manner. Such interactions are of particular interest especially when uncharacterized pathogens strike in a naïve population.

One such pathogen is the Zika virus, a positive stranded RNA virus belonging to the *Flaviviridae* family, causing the first large unexpected outbreak in Asia and French Polynesia in 2013 [1]. In 2016, a sudden raise in incidence rates of microcephaly in newborns in southern America could be linked to the ongoing Zika virus (ZIKV) outbreak in this otherwise naïve population [2,3]. In adults, ZIKV is predominantly asymptomatic or causes mild, flu-like symptoms, although recent studies report visual impairment and infertility [3,4,5]. Following vertical transmission upon maternal infection, ZIKV can cause fetal demise or Guillain-Barré syndrome, a neuro-inflammatory disease of the peripheral nervous system [6,7]. However, the limited knowledge on ZIKV pathogenesis and the sudden spread of this pathogen sparked a number of independent laboratories to characterize the virus–host interactions on a molecular level. These studies are of particular interest, since ZIKV combines many unique features of flaviviruses that are normally limited to individual species. Indeed, similarly to other members of the *Flaviviridae* family like Dengue (DENV), Yellow fever (YFV), Japanese encephalitis virus (JEV) and West Nile virus (WNV), ZIKV is also transmitted by mosquitoes. Furthermore, analogously to JEV, WNV and tick-borne encephalitis virus (TBEV), Zika virus infections are associated with neurological diseases, although ZIKV is the only *Flaviviridae* member with documented ability to cross the placental barrier and be sexually transmitted [8].

Technical advancement in mass-spectrometry analysis and the availability of replication competent epitope-tagged viruses and individual viral proteins have prompted several studies to investigate persistent modulation of protein abundances, post-translational modifications and protein–protein interactions on a global scale and with unprecedented sensitivity and quantitation [9,10,11,12,13]. Given its unique properties and the urgent need to further understand ZIKV biology, a number of laboratories started to use large-scale approaches to study how this pathogen interacts with cellular proteins. Here we review current approaches, in particular proteomic-based datasets, which have been used to identify ZIKV–host interactions.

## 2. Illuminating Dark Matter: Unbiased Approaches to Study Virus–Host Interactions

The recent technological revolution allows acquiring knowledge on biological processes at incredible speed. Several systematic and unbiased genetic approaches to identify host factors involved in flavivirus replication have been conducted to date [14,15,16,17,18,19,20,21]. For instance, next generation sequencing (NGS) became an affordable and widely used technology to unearth underrepresented sequences or estimate the abundance of RNA transcripts in complex mixtures. This made possible the identification of newly emerging viruses and virus variants. Illustrative examples in this direction are efforts during public health responses as recently shown by ZIKV and Ebola virus epidemics [22,23,24]. Furthermore, NGS can be used to study widespread transcriptional changes occurring upon virus infection, including expression of cytokines and antiviral response patterns. Deconvolution of such data using bioinformatics techniques offers a global overview of transcriptional responses and allows identifying viral activities to perturb signaling cascades. However, viruses modulate gene expression in a post-transcriptional manner by affecting mRNA processing, transport, translation, protein maturation and stability and therefore information on mRNA abundance is only partially reflecting the expression on protein level [25]. Mass spectrometry (MS)-based proteomics evolved into a sensitive and reliable technique that allows identifying proteins in complex mixtures [26]. Comparisons between mRNA and protein expression patterns in virus-infected cells revealed dramatic differences, indicating that protein abundance does not necessarily follow expression levels on transcript level [27]. Labeling proteins of a subcellular compartment (e.g., the cell surface) or combining fractionation techniques with mass spectrometry, allows assigning subcellular localization patterns to individual proteins and also monitor their translocation between compartments in a systematic manner [28,29]. MS can also identify post-translational modifications (PTMs) such as differential phosphorylation, ubiquitination or acetylation patterns. Such approaches, however, often require elaborative enrichment of modified peptides and mostly provide descriptive insights into PTMs, and functional consequences of individual modifications remain often elusive. In addition to alterations in protein expression patterns and protein quality, MS puts proteins into context. Isolation of protein complexes through affinity purification (AP) followed by MS analysis (AP-MS) gives insights into protein–protein interactions and engagement of cellular machineries. AP-MS has successfully been used to identify protein complexes and can put proteins in functional context (Figure 1).

Development of sample preparation methodologies, highly sensitive mass analyzers as well as tailored bioinformatics processing pipelines (comprehensively summarized in [30]) have caused MS to flourish in the last years. MS is no longer a niche technology reserved to a limited number of experts but is now available for a wide range of scientists interested in biological sciences. This, however, also led to generation of datasets that are exceedingly difficult to compare and to interpret due to the complexity of the underlying analysis and the wealth of data that is generated. Here we review current literature on proteomic approaches that were employed to study flaviviruses.

## 3. Flavivirus Interactions with the Cellular Proteome

Some members of the *Flaviviridae* family are considered highly pathogenic infectious agents and knowledge on their interactions is pivotal to understand these pathogens in order to identify new forms of therapy. This is particularly true for epidemic viruses such as Zika virus, which since 2016 has left a devastating impact on large populations in southern America. In an effort to understand the pathogenicity of this virus, a number of laboratories employed systems analysis to elucidate the functions of individual proteins and the virus as a whole.

### 3.1. ZIKV–Host Interactions

#### 3.1.1. Interactions of ZIKV Capsid with Host Proteins

Systematic studies are of paramount importance to identify virus–host protein–protein interactions (PPIs), since they provide an unbiased view of molecular complexes and therefore insights into viral mechanisms of replication, host resources exploitation and immune evasion strategies [11,12,31,32]. Applying AP-MS in a systematic and comparative fashion can help elucidate similarities and differences in virus-specific replication strategies and even pathogenesis determinants [9,33]. A few studies recently reported combined proteomic and gene perturbation approaches integrating complementary methods to investigate the cellular interactome of each of the 10 ZIKV proteins (Figure 2a). A discrete number of previously reported bona fide interactors were consistently identified across all these studies, and appear to be shared across different flavivirus members, highlighting the relatively high degree of conservation within the genus and across diverse cellular backgrounds, thereby revealing cellular components critically required for virus replication. However, these studies also revealed specificities, reflecting the different methodologies, cellular backgrounds, enrichment strategies, gene-delivery methods and experimental designs used. For instance, using an AP-LC-MS/MS (affinity purification coupled with liquid chromatography and tandem mass spectrometry) approach in SK-N-BE2 neuroblastoma cells [34], we identified in the ZIKV capsid interactome a strong enrichment in nuclear and nucleolar-resident proteins such as nucleolin (NCL), nucleolar RNA-dependent helicases of the DDX family, core histones (H2A), as well as peroxisomal proteins including Pex19, that have been previously reported as cellular targets of Dengue (DENV) and West Nile virus (WNV) (Figure 2b) [35,36,37]. Interestingly, among the capsid-specific interactors, a completely new set of proteins associating with the ZIKV capsid was identified. Among these, a poorly characterized nucleolar protein involved in cell-growth regulation and maintenance of stem cell identity called LYAR was identified [38]. LYAR was recently implicated in viral transcription and replication of Influenza A virus (IAV) through interaction with the viral ribonucleoprotein [39], indicating a more global role of this protein in virus–host interactions. Additionally, several members of the LARP family (e.g., LARP1 and LARP7) and ZC3HAV1 (also known as ZAP) were identified as specific capsid interactors, suggesting the propensity of ZIKV capsid to interact with RNA-binding proteins. Importantly the functional relevance of some of these ZIKV capsid interactors was underlined by knock-down studies, confirming a strong reduction of viral replication upon gene depletion. In similar studies, Coyaud and collegues [40] as well as Shah et al. [41] confirmed these observations, identifying NCL, LYAR, LARP1, several members of the DDX family and ZC3HAV1, among the specific ZIKV capsid interactors using BioID- and AP-LC-MS/MS in 293T cells, respectively. Furthermore, recent work by Li et al. reported analogous enrichment for LARP1, LARP7 and ZC3HAV1 in a global proteomic survey for WNV-interacting proteins [42]. Complementary functional experiments validated a conserved role for ZC3HAV1 in restricting flaviviruses since depletion of ZC3HAV1 led to a 4- to 8-fold increase in ZIKV, DENV and WNV virus titers. Interestingly, the antiviral activity of ZC3HAV1 has been investigated for several virus families including human immunodeficiency virus (HIV) and Sindbis virus (SINV) [43,44]. The protein appears to recruit cellular RNA degradation machineries to specific ZAP-responsive elements on the viral mRNAs as well as the DCP1–DCP2 complex to initiate 5′-3′ RNA degradation of viral mRNA. Its activity has been mechanistically studied in the context of Japanese encephalitis virus, whereby ZC3HAV1 has been described to recruit the 3′-5′ degradation machinery [45]. However, the exact mechanisms of virus restriction by ZC3HAV1 remain to be elucidated.

Among the other novel capsid interactors, additional work by the Ott and the Ramage groups identified the nonsense-mediated decay (NMD) pathway and the exon-junction complex (EJC) as targets of the WNV, DENV and ZIKA virus capsids [42,46]. While the antiviral activity of individual members of these complexes appears moderate in knock-down studies, further experimental evidence supports a direct interaction between the viral RNA and RBM8A (a central component of the EJC), suggesting a depletion of the cellular PYM1 pool upon recruitment by the WNV capsid to protect viral RNA from decay. Similarly, direct involvement of the NMD pathway in flavivirus is corroborated by the identification of the regulator of nonsense transcripts 1 (UPF1) among the ZIKV capsid interactors and the upregulation of several NMD substrate mRNAs upon ZIKV infection both in Huh7 and hNPC. Furthermore, depletion of UPF1 leads to a 50% increase in virus replication, supporting the notion that NMD acts as flavivirus restriction machinery that is counteracted by viral activities [46].

In addition to proteins recruited by several flavivirus capsids, the ZIKV capsid is the only one reported to specifically associate with cellular factors that are involved in neuronal development or neurological disorders [34]. These proteins include neuroguidin (NGDN), an eIF4E-interacting protein mediating CPEB-mediated translation of essential genes in early development and neuronal synaptic plasticity. Further ZIKA capsid interactors are several members of the survival motor neuron (SMN) complex [34], which is important for proper splicing, neuronal migration and differentiation. It would be interesting to assess binding specificity of other flavivirus capsid proteins in a similar neuronal background and investigate further their contribution to viral replication or pathogenesis. Such studies could pinpoint novel viral pathogenesis determinants and assist in the identification of druggable binding surfaces required for specific protein–protein interactions.

Altogether, flavivirus capsids evidently evolved evasive mechanisms through binding/sequestration of ZC3HAV1 and other RNA-binding proteins (e.g., UPF1, LARP7, NGDN). Capsid proteins are apparently exploiting their versatile subcellular localization (both nuclear and cytoplasmic) and their intrinsic hydrophobic character to simultaneously highjack and diverge regulation or expression of specific transcripts.

#### 3.1.2. Interactions of ZIKV NS4B with the Host

Another interesting ZIKV protein that appears to have both conserved and unique cellular interactors across flaviviruses and plays a central role in virus–host adaptation, is the non-structural protein NS4B. Indeed, ZIKV-NS4B associates with subunits of the ATP synthase (ATP1A1, 1A2, 1A3, 1B and 6V1H), voltage-dependent anion selective channel proteins (VDAC1,2,3) and calcium-binding mitochondrial proteins (e.g., SLC25A, SLC38A1), as well as several components of the cytochrome C oxidase complex (COX15, MT-CO2, NDUFA4) (Figure 2c). These results largely mirror those of an earlier study on the DENV-NS4B interactome in Huh7 hepatoma cells [47], confirming ATP production, calcium homeostasis, apoptosis, and mitochondrial respiratory chain, as important targets of different flaviviruses. These interactors argue for a conserved association between flavivirus NS4Bs and mitochondria in virus-infected cells, eventually leading to a profound alteration of mitochondria morphodynamics. Since mitochondria are central to regulate antiviral immune responses, these interactions may contribute to perturbation of innate immune mechanisms through mitochondria elongation as proposed recently [47]. In addition to mitochondrial proteins, ZIKV NS4B appears to bind cellular proteins involved in protein stability and quality control in both SK-N-BE2 and HEK293T cells. For instance, STT3B, is involved in endoplasmic reticulum-associated degradation, ICMT, mediating targeting of isoprenylated proteins to the cell membrane, and the autophagosome-associated protein SQSTM1 (p62), which is in line with a general and conserved role of autophagy in flavivirus infections [34,41,48,49,50].

NS4B also appeared to associate with cellular proteins that could be associated with the whole spectrum of ZIKV-associated pathogenesis including neurodegenerative disorders and retinal degeneration (CLN6, BSG), neuronal differentiation defects (CEND1, RBFOX2) and axonal dysfunction (CHP1, TMEM41b) [34]. While these cellular factors appeared as specific binders of arthropod-borne flaviviruses in neuronal cells (such as DENV and ZIKV), a subset of proteins exhibited ZIKV specificity (TMEM41b, CEND1, CLN6), suggesting that similarly to capsid, also NS4B might have evolved partially divergent binding affinities across different flaviviruses, with potential consequences for the distinctive pathogenic outcome. Interestingly CLN6, a poorly characterized protein associated with a neurodegenerative disease with late-infantile onset, shares several cellular binding partners with NS4B, and specifically associates with mTOR. These results support reports associating ZIKV infection or NS4B overexpression to defective neurogenesis via suppression of the AKT-mTOR signaling pathway [51]. Another conserved target of multiple flavivirus NS4Bs is the translocon complex, with the Sec61A and Sec61B subunits reported to bind NS4A [41] or NS2B3 and NS4B [34]. The critical requirement of this complex for flavivirus replication both in mammalian and insect cells was further supported by the potent antiviral activity of CT8, a specific translocon inhibitor [21,52] as well as its intracellular accumulation in NS3/NS4B-positive convoluted membranes in DENV-infected cells [47].

#### 3.1.3. Other Viral Proteins

In addition to capsid and NS4B, several novel ZIKV NS5 binders were recently identified in these proteomics surveys. Among these, multiple members of the PAF1C complex (Leo1, CDC73, CTR9 and PAF1), confirming similar observations previously made on DENV-NS5 by the Gamarnik group [53] and recently confirmed by others [54]. Importantly, while these interactions appear highly conserved among DENV, ZIKV, JEV and WNV-NS5s, a moderate and reciprocal effect on viral replication was observed upon gene silencing (2-fold increase and 2-fold decrease for ZIKV and JEV infectivity, respectively) [41,54]. Further mechanistic studies suggest that PAF1 might be recruited by NS5 to reduce expression of interferon-stimulated genes and therefore dampen immune responses [41]. Interestingly, NS5 has also been associated with modulation of alternative splicing, as reported by earlier studies on the DENV-NS5 interactome revealing an association of NS5 with the U5 snRNP particle, CD2BP2 and DDX23, providing evidence of viral interference with alternative splicing events, eventually leading to changes in the abundance of specific mRNA isoforms of known antiviral factors [53].

Furthermore, among the novel cellular targets identified in the ZIKV NS4A interactome is ANKLE2, a protein previously shown to cause autosomal recessive microcephaly in humans [41]. Interestingly, in an ANKLE2-heterozygous mutant drosophila model, expression of NS4A reduced brain size, suggesting that NS4A might induce neurotoxic effects, at least under limiting amounts of functional ANKLE2.

An important aspect is that AP-LC-MS/MS often identifies protein complexes rather than binary protein–protein interactions. This is also illustrated in an interactome study of DENV-NS1 [55] in the context of a fully replicating virus. The authors employed three different cell lines (Raji, Hela and HAP1) to exploit similarities and specificities of *bona fide* interactors in diverse cellular backgrounds as a selection criterion to pinpoint targets relevant for *in vivo* infections. This study identified a strong enrichment of the CCT complex (cytosolic chaperonin-containing T complex), and subunits of the OST (oligosaccharyl transferase) complex such as DDOST, STT3A, STT3B and RPN2, in the NS1 interactome. Previous reports identified some of these proteins as host-dependency or druggable targets in genetic [14,56] and drug-based screens [57], and the interactome study now provides potential mechanistic insights into the underlying mechanisms of anti-viral activity. However, in light of cumulative evidence on ZIKV and WNV, it is tempting to speculate that this interaction might also require additional viral binding partners (e.g., NS4A and NS4B) or a productive virus infection, since ectopic expression of NS1 does not recapitulate this interaction [13,34,40,41,42]. In this respect, mass spectrometry has also been instrumental to identify interactions between viral proteins and that could explain some of their cooperative functions. NS1 of WNV, for instance, interacts with NS4B (WNV, [58]). Likewise, DENV NS1 associates with an unprocessed viral precursor NS4A-2k-NS4B (DENV, [59]) and NS4B of DENV has been reported to bind NS3 (DENV, [60]). Proteomic analysis, in addition to identifying functional protein complexes, fosters identification of yet unreported viral gene products (e.g., NS4A-2K-NS4B). Such information is helpful to extend our view on the function of viral proteins and the mode of action of therapeutic drugs.

#### 3.1.4. Interactions of ZIKV Viral RNA with the Host

In addition to PPI, flaviviruses were also shown to exploit their own genomic RNA (gRNA) or subgenomic derivatives, such as the subgenomic flavivirus RNA (sfRNA), to “sponge” critical RNA-binding host proteins out of the cellular pool, or to specifically inhibit or redirect their homeostatic regulatory functions [61,62,63]. In this respect, recent studies have tried to chart systematically these interactions using different MS-based approaches. For instance, using a TUX-MS-based method (thiouracil cross-linking mass-spectrometry), Viktorovskaya and colleagues reported the identification of 79 proteins specifically associated with the DENV gRNA [64]. Among these, proteins earlier involved in PPI or associated with flavivirus replication were consistently identified (e.g., PAF1 and multiple hnRNPs), as well as novel host factors previously not associated with flavivirus infections (e.g., RBMX, RNA-binding motif protein X chromosome; NONO, non-POU domain-containing octamer-binding protein; and HMCES, embryonic stem cell-specific 5-hydroxymethylcytosine-binding protein). Importantly, depletion of RBMX and NONO by RNAi significantly reduced DENV titers, confirming their functional relevance in DENV replication. Similar studies were also performed using an alternative cross-linking approach based on the use of 5′-end biotinylated antisense oligonucleotides specific to the DENV gRNA and very stringent cut-off criteria. Interestingly this approach identified only 12 host proteins specifically associated with the gRNA and further validated as bona fide interactors in RNA-IP-WB experiments. Among these, depletion of CSDE1 (cold-shock domain-containing protein E1) and hnRNPC (heterogeneous nuclear ribonucleoproteins C1/C2) reduced ~50% of infectious titers [65].

Only one study to date reported the systematic analysis of ZIKV and DENV gRNA-associated proteins, using the ChIRP-MS (comprehensive identification of RNA-binding proteins by mass spectrometry) method. In this method, target RNA and associated proteins are retrieved after formaldehyde-based cross-linking using hybridization with antisense biotinylated oligos and streptavidin-conjugated beads, and protein complexes eluted via incubation with biotin [66]. This approach identified 464 RNA-binding proteins (RBPs) associated with DENV or ZIKV gRNAs [67], confirming association with previously reported flavivirus RBPs (e.g., YBX1, MOV10 and ADAR) and host-dependency factors (e.g., STT3B and MAGT1). Importantly, intersection of these large datasets with previously published genome-wide CRISPR/Cas9-based screens, as well as four newly performed surveys using all four DENV serotypes, identified Vigilin and RRBP1 (ribosome-binding protein 1) as novel RBPs not previously linked to flavivirus infection. Interestingly, both of these proteins promote DENV infection and gRNA stability, positively modulating translation and replication of the gRNA.

### 3.2. Global Proteome Expression Affected by ZIKV Infection

Several studies systematically investigated the impact of flavivirus infections on protein abundance on a global scale [68,69]. Garcez and colleagues were the first to use label-free shotgun proteomics combined with transcriptomic analysis to study the impact of a Brazilian isolate of ZIKV on human neurospheres [70]. This approach revealed 199 downregulated proteins and 259 upregulated proteins, including proteins with functions in arrest of cell cycle progression (e.g., CDK2, ERBB2) and neural differentiation (e.g., HDAC7 and NeuroD1) or previously associated with viral replication (e.g., MAP4K4, TLR4 and DDX6). Importantly, this approach also revealed an activation of the DNA repair machinery, as inferred by an upregulation of proteins such as FANCD1, BRCA1 or MRE11A. Interestingly, these observations were later complemented by the identification of several hyperphosphorylation events across the DNA damage signaling pathway (see Section 3.1), as well as recent independent reports describing induction of double-strand DNA breaks (DSBs) and activation of DNA damage response in ZIKV-infected hNPC [71].

A similar unbiased study investigating the proteome of ZIKV-infected iPSC-derived human neuronal progenitors, confirmed specific downregulation of neuronal factors upon ZIKV infection, as well as activation of general antiviral response pathways, such as upregulation of type-I interferon stimulated genes (e.g., STAT1, MX1, OAS3, IFIT1) [34]. Interestingly, this approach was also used to investigate early changes occurring in ZIKV-infected or NS4B-transduced hNPC at the onset of neuronal differentiation, revealing specific downregulation of factors involved in neuronal differentiation as well as proteins associated with neurological diseases, such as DOK3 and SUMO2, thereby revealing potential targets deregulated during neurogenesis. Importantly, in line with previous studies pointing to a role of NS4A and NS4B in inhibition of neurogenesis and deregulation of the Akt-mTOR signaling pathway [51], this study also identified shared targets deregulated both by ZIKV infection and NS4B expression (e.g., MAP2, -6, DPYSL3, -5, CNTN2).

Recently, additional reports investigated further the impact of ZIKV infection on human mesenchymal stem cells using the MudPIT method (multidimensional protein identification technology). This identified a profound downregulation of PI3K/AKT and mTOR, as well as dysregulation of several components of the phosphatidylinositol pathway [72]. Alternative studies using shotgun proteomics and TMT (tandem mass tag) labeling confirmed and expanded the observed impairment of neurogenesis and synaptogenesis previously observed in 2D- and 3D-models of ZIKV-infected NPC, also to neurons and neural stem cells [73]. Using this method to systematically compare African and Asian ZIKV isolate representatives, revealed differential responses especially in the context of neurospheres. Among these, HSPB1 emerged as one of the most downregulated proteins in neurospheres, while DCX was profoundly upregulated in NPC, confirming the existence of significant viral isolate- and cellular background-dependent specificities on proteins that are relevant for neuron-specific functions.

A unifying theme crystallizing from proteomic and transcriptomic analysis of ZIKA virus infected neuronal cells is that ZIKV infection has a profound effect on expression of proteins involved in neurogenesis or on proteins that are markers of neuronal differentiation. At this stage it is not clear whether this effect is limited to ZIKA virus or whether other viruses elicit similar effects in a neuronal background. Along these lines would be important to test whether neuropathic flaviviruses share certain binding patterns that may be etiologically linked to their pathology. Similarly, it would be of great interest to investigate the differential proteomes of diverse flaviviruses from primary biopsies of infected patients.

### 3.3. Phosphorylation and Other PTMs Modulated by ZIKV

Initial studies by the Cao group focused on the global protein phosphorylation status in WNV- and JEV-infected cells [74,75]. In the case of WNV, 1657 proteins were found as significantly regulated at the phosphorylation level, and gene ontology enrichment analysis identified ErbB, NFκB and mTOR signaling pathways as the strongest deregulated pathways. Furthermore, several important kinases and substrates such as RNKP, RB1 and GSK3B were hyperphosphorylated, and depletion of these proteins by RNA interference significantly reduced the level of inflammatory cytokines such as IL-6, IL-1β and TNF-α in WNV-infected cells. Similarly, analysis of the cellular phosphoproteome upon JEV infection identified over 1200 differentially modulated phosphoproteins, including AKT, GSK-3β and PKC as well as ERK and p53. Interestingly, in case of JEV, a specific enrichment of JNK1 and CK2 substrates was identified among hyperphosphorylated and hypophosporylated proteins, respectively. Importantly, pharmacological modulation of JNK1 signaling reduced JEV-induced production of inflammatory cytokines in the brain of infected mice, suggesting an important role for stress-mediated responses in JEV-mediated neurotoxicity [74].

In the case of ZIKV, only one study so far reported a time-resolved phosphoproteomic analysis of ZIKV-infected cells [34]. Similarly, to JEV and WNV, 1216 proteins were modulated upon ZIKV infection, and analysis of enriched cellular functions affected by ZIKV infection identified cellular assembly and organization, cell-cycle regulation, as well as nervous system (NS) development and neurological diseases. In line with earlier reports, also in case of ZIKV the ATM, AKT/mTOR and ERK/MAPK signaling cascades were significantly regulated, as inferred by dephosphorylation of multiple AKT1 substrates (e.g., DNMT1, TBC1D4, LARP6), mTOR targets (e.g., ANKRD17, LARP1, BAG3, EEF2K, WNK1), the central kinase RPS6KB1 and its main effector protein RPS6. Similarly, several downstream substrates of the ERK1/2 map kinases (e.g., EIF4EBP1, BAZ1B, TWIST) were significantly dephosphorylated upon ZIKV infection. In this respect, these data complement on a different level of earlier observations reporting a downregulation of the AKT/mTOR pathway in ZIKV-infected hNPC and neurospheres [51].

Furthermore, a profound upregulation of the ATM (ataxia-telangiectasia mutated)/ATR (ATM- and Rad3-related) DNA-damage pathway, was observed at multiple levels (e.g., hyperphosphorylation of several substrates of ATR, DNAPK and downstream effector proteins) including a number of proteins involved in cell-cycle regulation and DNA damage checkpoint (TOP2B, CDK1). Altogether, these effects provide possible explanations for the observed ZIKV-induced cell-cycle arrest and apoptosis [76] and p53 activation [77] and are in line with multiple other reports describing persistent activation of DNA damage repair pathways in response to ZIKV-induced mitotic abnormalities [76]. Furthermore, these effects appear to be ZIKV-specific, at least when compared to DENV infection in hNPC [71], suggesting an important role in cell death of neuronal progenitors. Phosphoproteomic analysis also supports a prominent role of Zika virus on neurogenesis. p38 MAPK and downstream targets (HSPBI and ATF7), MARCKS (one of the main PKC substrates) and DPSYL2, all of which positively modulate neurite outgrowth and brain development [78,79,80] are differentially phosphorylated in cells infected with ZIKA virus. However, at this stage only a relatively limited number of phosphorylation events can be linked to a given function in a protein. In light of the abundance of differentially phosphorylated residues that have not been reported previously, such datasets constitute a largely unexplored treasure that will reveal its enormous potential only in light of mechanistic functional studies on individual phosphorylation sites.

Another important PTM is ubiquitination, which becomes more and more accessible to proteomics analysis and has recently been studied in the context of flavivirus infections [81]. For instance, Zhang and colleagues recently reported the use of an elegant hydrolysis-resistant ubiquitin variant (HA-UbL73*) in a functional proteomic screen. This work identified several host proteins differentially mono-ubiquitylated upon DENV infection, unraveling the ER-resident protein AUP1 as differentially ubiquitin-modified in DENV-infected cells. AUP1, a lipid droplet-localized type-III membrane protein existing predominantly in the mono-ubiquitylated form, appears to be the main trigger of DENV-induced lipophagy, and the newly identified NS4A–AUP1 interaction is sufficient to trigger its acyltransferase activity.

Collectively, the analysis of PTMs in flavivirus-infected cells starts to unravel unappreciated cellular activities. Development of techniques to study PTMs as well as development in bioinformatic tools allowing better interpretation of PTM activities, hold promise to dramatically accelerate our understanding of virus–host interactions and to appreciate additional important pathways engaged during virus infections.

## 4. Conclusions and Future Perspectives

Large-scale approaches allowing the accurate assessment of virus–host interactions have revolutionized the way we investigate and deconvolute the complex cellular changes triggered by virus infections. While central linear pathways were the main research focus in the past, it becomes more evident that innate immune signaling, and cellular responses in general, engage many interconnected pathways in parallel. The knowledge of these pathways is paramount to holistically understand how viral pathogens interact with their hosts. Certainly more work on viral–host interactions, protein turnover as well as post-translational modifications is required to further expand our understanding of the intracellular landscape of flavivirus-infected cells. This requires well-controlled comparative studies in relevant cell culture systems of multiple members of this highly diverse arbovirus family. Similarly, rigorous validation methods by alternative approaches (e.g., co-IP-WB or proximity ligation assays), robust and transparent cut-off criteria, and functional assays on selected candidates are increasingly needed to pinpoint physiologically relevant host factors. In this respect, further efforts are required to integrate and standardize data generated by mass spectrometry based surveys in easy-to-mine databases, ideally providing critical annotations on virus isolates and cell culture systems as well as integration with other -omics data [82]. Integrative efforts to merge such information gathered at different levels into organic maps of virus-induced perturbations will be imperative to fully exploit their enormous translational potential.

## Figures and Tables

**Figure 1 viruses-11-00878-f001:**
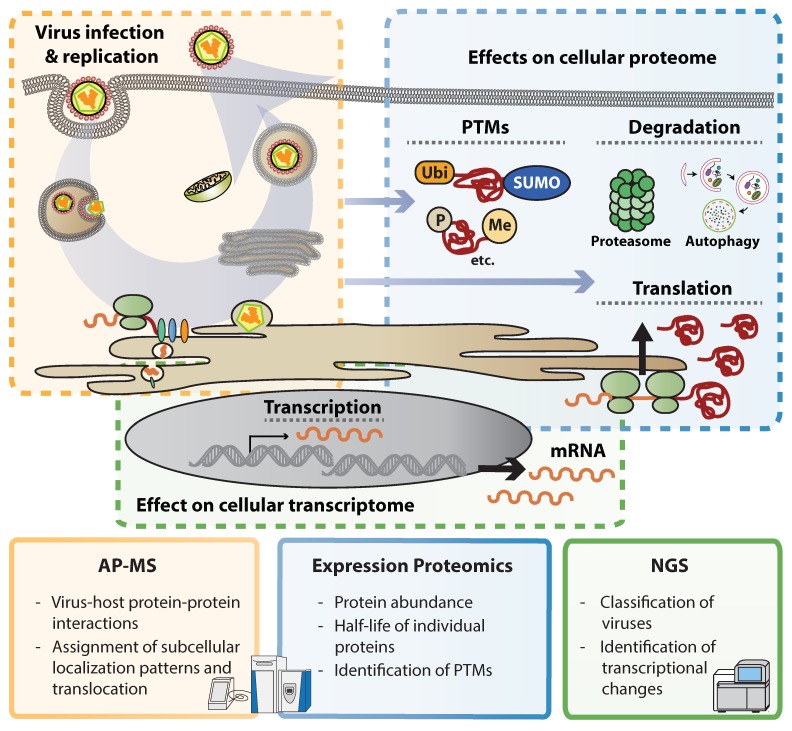
– Omics methods to study virus-induced cellular perturbations. The key advantages of Interaction proteomics (affinity purification mass spectrometry, AP-MS), expression proteomics and next generation sequencing (NGS) in studying flavivirus-induced perturbations are listed within each square. Colors indicate the respective step(s) of the viral replication cycles predominantly targeted by each method. Abbreviations: PTMs, post-translational modifications.

**Figure 2 viruses-11-00878-f002:**
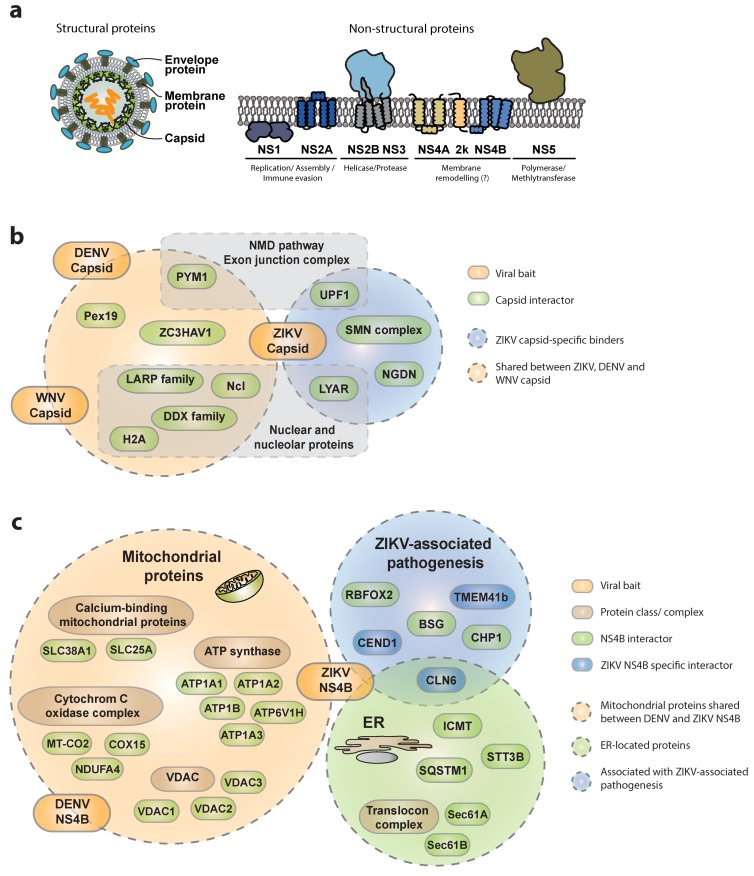
– Recently identified cellular interactors of flavivirus capsids and NS4Bs. (**a**) Schematic representation of flavivirus viral proteins and respective membrane topology. (**b**) Capsid interactors reported to bind exclusively to Zika virus (ZIKV)-C are surrounded by the light blue circle, while those within the orange circle were reported as shared interactors of ZIKV, West Nile virus (WNV) and Dengue (DENV)-C. The respective sub-cellular localization is indicated. (**c**) NS4B interactors reported to bind exclusively to ZIKV-NS4B are shown in blue, while those shared with DENV-NS4B are shown in green. Mitochondrial- and endoplasmic-reticulum-resident proteins are surrounded by green and orange circles, respectively. Proteins with functions associated to ZIKV-related pathologies are surrounded by the blue circle. All gene names and corresponding full protein names are listed in Table 1.

**Table 1 viruses-11-00878-t001:** Gene names and corresponding full protein names listed in alphabetical order.

Gene Name	Protein Name	Gene Name	Protein Name
AKT	RAC-alpha serine/threonine-protein kinase	ICMT	protein-S-isoprenylcysteine O-methyltransferase
ANKLE2	Ankyrin repeat and LEM domain-containing protein 2	LARP	La-related protein
ANKRD17	ankyrin repeat domain-containing protein 17	LEO1	RNA polymerase-associated protein 1
ATF7	cyclic AMP-dependent transcription factor ATF-7	LYAR	cell growth-regulating nucleolar protein
AUP1	ancient ubiquitous protein 1	MAP4K4	mitogen-activated protein kinase kinase kinase kinase 4
BAG3	BAG family molecular chaperone regulator 3	MARCKS	myristoylated alanine-rich C-kinase substrate
BAZ1B	tyrosine-protein kinase BAZ1B	MRE11A	double-strand break repair protein MRE11
BRCA1	breast cancer type 1 susceptibility protein	MT-CO2	cytochrome c oxidaes subunit 2
CD2BP2	CD2 antigen cytoplasmic tail-binding protein 2	mTOR	mechanistic target of rapamycin
CDC73	cell division control protein 73	NCL	nucleolin
CDK1	cyclin-dependent kinase 1	NDUFA4	cytochrome c oxidase subunit NDUFA4
CDK2	cyclin-dependent kinase 2	NeuroD1	neurogenic differentiation factor 1
CEND1	cell-cycle exit and neuronal differentiation protein 1	NGDN	neuroguidin
CLN6	ceroid-lipofuscinosis neuronal protein 6	NONO	non-POU domain-containing octamer-binding protein
COX15	cytochrome c oxidase assembly protein COX15 homolog	PAF1	RNA polymerase II-associated factor homolog
CPEB	cytoplasmic polyadenylation element binding protein	PEX19	peroxisomal biogensis factor 19
CSDE1	cold-shock domain-containing protein E1	PYM1	partner of Y14 and mago
CTR9	RNA polymerase-associated protein CTR9 homolog	RBM8A	RNA-binding protein 8A
DCP1-DCP2	mRNA-decapping enzyme subunit 1-2rbm8A	RBMX	RNA-binding motif protein X chromosome
DCX	neuronal migration protein doublecortin	RPN2	dolichyl-diphosphooligosaccharide-protein glycosyltransferase subunit 2
DDOST	dolichyl-diphosphooligosaccharide-protein glycosyltransferase 48 kDa subunit	RRBP1	ribosome-binding protein 1
DDX	ATP-dependent RNA helicase	SMN1	survival motor neuron protein 1
DNMT1	DNA (cytosine-5)-methyltransferase 1	STT3A/B	dolichyl-diphosphooligosaccharide--protein glycosyltransferase subunit A/B
DOK3	docking protein 3	TBC1D4	TBC1 domain family member 4
DPYSL2	dihydropyrimidinase-related protein 2	TLR4	toll-like receptor 4
EEF2K	eukaryotic elongation factor 2 kinase	TMEM41b	transmembrane protein 41beta
ERBB2	Receptor tyrosine-protein kinase erbB-2	TOP2B	DNA topoisomerase 2-beta
FANCD1	breast cancer type 2 susceptibility protein	TWIST	twist-related protein 1
HDAC7	histone deacetylase 7	UPF1	regulator of nonsense transcripts 1
HMCES	embryonic stem cell-specific 5-hydroxymethylcytosine-binding protein	VDAC1, 2, 3	voltage-dependent anion selective channel protein 1, 2, 3
hnRNPC	heterogeneous nuclear ribonucleoproteins C1/C2	WNK1	Serine/threonine-protein kinase WNK1
HSPB1	heat shock protein beta-1	ZC3HAV1	zinc finger CCCH-type antiviral protein 1

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
