# Peer review of "Chasing Intracellular Zika Virus Using Proteomics"

_viruses, 2019, doi:10.3390/v11090878_

Round 1

Reviewer 1 Report

In the manuscript 'Chasing intracellular ZIKA virus using proteomics', the authors provide a review of mass-spectrometry-centric proteomics studies in flavivirus biology. As stated in the title, the authors chose Zika virus as an example of virus-host interactions.

My general impression is that the manuscript is well written and provides a thorough summary of the current status quo in mass-spectrometry-based flavivirus-host protein-protein interaction research. There are, however, a few issues I would like to address before publication.

My major point of criticism relates to the fact that the authors focus on protein-protein interactions, thereby completely neglecting RNA-protein interactions. The latter are well studied and the their role in flavivirus pathology is well established in literature (see e.g. doi:10.3390/v6020404 , doi:10.1126/science.aam9243 , doi:10.1038/s41598-019-43390-5 , doi:10.1016/j.coviro.2014.09.001 ). I'm sure there are proteomics studies focusing on RNA-binding proteins. The authors might consider adding a section discussing those as well.

Another topic I would appreciate to see in this manuscript is a short review (not more than one or two paragraphs) of current bioinformatics approaches in virus proteomics. While I am aware that this is not in the main focus of the present manuscript, I think it would augment the overall picture.

Author Response

In the manuscript 'Chasing intracellular ZIKA virus using proteomics', the authors provide a review of mass-spectrometry-centric proteomics studies in flavivirus biology. As stated in the title, the authors chose Zika virus as an example of virus-host interactions.

My general impression is that the manuscript is well written and provides a thorough summary of the current status quo in mass-spectrometry-based flavivirus-host protein-protein interaction research. There are, however, a few issues I would like to address before publication.

My major point of criticism relates to the fact that the authors focus on protein-protein interactions, thereby completely neglecting RNA-protein interactions. The latter are well studied and the their role in flavivirus pathology is well established in literature (see e.g. doi:10.3390/v6020404 , doi:10.1126/science.aam9243 , doi:10.1038/s41598-019-43390-5 , doi:10.1016/j.coviro.2014.09.001 ). I'm sure there are proteomics studies focusing on RNA-binding proteins. The authors might consider adding a section discussing those as well.

We completely agree with the reviewer, and have now included some of the suggested references, and added a new sub-section summarizing the main findings of MS-based RNA-protein interaction surveys.

Another topic I would appreciate to see in this manuscript is a short review (not more than one or two paragraphs) of current bioinformatics approaches in virus proteomics. While I am aware that this is not in the main focus of the present manuscript, I think it would augment the overall picture.

We agree that bioinformatics approaches are of great interest for the readers and highly important for the MS-based virus systems biology field. However, we feel that any attempt to summarize the bioinformatics approaches in few sentences would oversimplify the state-of-the-art and could not reflect the diverse methods that are around to analyse MS data. In the revised version of the manuscript we now refer to an excellent review that is dedicated to bioinformatics analysis of MS data and that addresses this point thoroughly. We ask reviewer #1 for his understanding and his consent to omit a section on bioinformatics analysis.

Reviewer 2 Report

A well written review of the current progress and achievements on proteomics for flaviviruses, with a specific emphasis on ZIKV.

Minor comments

1. A diagram or description for gene products of flaviviruses is needed in the Introduction. Without this information, the readers would not fully understand the specific proteins that are play a role in virus-host interaction at the level of protein.

2. Line 76 > needs revision. e.g., viruses modulate

3. Line 134> needs rewriting . no verb. 

4. There are many gene names and protein names used in the manuscript; however, spelling out of gene names is inconsistent through out the manuscript. It would be better to have a table or appendix to list all genes listed here with a proper spelled out gene names. For example, what is LYAR?

5. Line 166 “,” should be removed.

6. There are various approaches to perform proteomics for protein-protein interaction: MolID, expression of a target gene in the target cells followed by IP-MS, and infection with actual viruses. The method that have been used for each referenced papers should be discussed in the respect of intrinsic limitation for each method. 

7. In Conclusion and the future direction section, a discussion on the validation of proteomics data would be appreciated by readers.

Author Response

A well-written review of the current progress and achievements on proteomics for flaviviruses, with a specific emphasis on ZIKV.

Minor comments

A diagram or description for gene products of flaviviruses is needed in the Introduction. Without this information, the readers would not fully understand the specific proteins that are play a role in virus-host interaction at the level of protein.       

We thank the reviewer for this helpful comment. We have now included a new schematic description of flavivirus polyprotein products in Figure 2 (panel A). 

Line 76 > needs revision. e.g., viruses modulate

            Done.

Line 134> needs rewriting . no verb.

            Corrected.

There are many gene names and protein names used in the manuscript; however, spelling out of gene names is inconsistent through out the manuscript. It would be better to have a table or appendix to list all genes listed here with a proper spelled out gene names. For example, what is LYAR?

This is an excellent idea, we have now included a table listing all the abbreviation and corresponding full gene names.

Line 166 “,” should be removed.

              Done.

There are various approaches to perform proteomics for protein-protein interaction: MolID, expression of a target gene in the target cells followed by IP-MS, and infection with actual viruses. The method that have been used for each referenced papers should be discussed in the respect of intrinsic limitation for each method.

We thank the reviewer for this comment. We agree that discussion of each method’s advantages and limitations would be of interest for the reader, but feel that would go beyond the scope of this review and focus of this review. We refer now to excellent reviews on the matter.

In Conclusion and the future direction section, a discussion on the validation of proteomics data would be appreciated by readers.

We have now added a few sentences in the Conclusions addressing validation of proteomics data.

Round 2

Reviewer 1 Report

All issues have been addressed by the authors